# Long-Term Circulation of Atypical Porcine Pestivirus (APPV) within Switzerland

**DOI:** 10.3390/v11070653

**Published:** 2019-07-17

**Authors:** Cindy Kaufmann, Hanspeter Stalder, Xaver Sidler, Sandra Renzullo, Corinne Gurtner, Alexander Grahofer, Matthias Schweizer

**Affiliations:** 1Institute of Virology and Immunology (IVI), 3001 Bern and 3147 Mittelhäusern, Switzerland; 2Department of Infectious Diseases and Pathobiology, Vetsuisse Faculty, University of Bern, 3001 Bern, Switzerland; 3Division of Swine Medicine, Department for Farm Animals, Vetsuisse Faculty, University of Zürich, 8057 Zürich, Switzerland; 4Institute of Animal Pathology, 3001 Bern, Switzerland; 5Clinic for Swine, Department of Clinical Veterinary Medicine, Vetsuisse Faculty, University of Bern, 3001 Bern, Switzerland

**Keywords:** atypical porcine pestivirus, APPV, epidemiology, prevalence, Switzerland, real-time RT-PCR, phylogenetic analysis, congenital tremor

## Abstract

In 2015, a new pestivirus was described in pig sera in the United States. This new “atypical porcine pestivirus” (APPV) was later associated with congenital tremor (CT) in newborn piglets. The virus appears to be distributed worldwide, but the limited knowledge of virus diversity and the use of various diagnostic tests prevent direct comparisons. Therefore, we developed an APPV-specific real-time RT-PCR assay in the 5′UTR of the viral genome to investigate both retro- and prospectively the strains present in Switzerland and their prevalence in domestic pigs. Overall, 1080 sera obtained between 1986 and 2018 were analyzed, revealing a virus prevalence of approximately 13% in pigs for slaughter, whereas it was less than 1% in breeding pigs. In the prospective study, APPV was also detected in piglets displaying CT. None of the samples could detect the Linda virus, which is another new pestivirus recently reported in Austria. Sequencing and phylogenetic analysis revealed a broad diversity of APP viruses in Switzerland that are considerably distinct from sequences reported from other isolates in Europe and overseas. This study indicates that APPV has already been widely circulating in Switzerland for many years, mainly in young animals, with 1986 being the earliest report of APPV worldwide.

## 1. Introduction

Myoclonia congenita, also known as congenital tremor (CT) or “dancing piglet”, is a disease occurring in neonatal piglets characterized by tonic–clonic contractions of the muscular skeletal apparatus. Constant shivering prevents the piglets from proper movement, causing problems in the nursing process, and leading to inadequate colostrum intake, starvation, and death. CT is classified into type A, which is associated with visible histological lesions in the affected tissues, and type B, which does not show any apparent lesions. Type A CT is further subclassified into five groups: type AIII and AIV are heritable forms in certain pig breeds and type AV is induced by intoxication with organophosphorus compounds during gestation, and both types, AI and AII, have infectious causes. Whereas type AI is caused by transplacental infection with classical swine fever virus (CSFV), the etiology of type AII was unclear for decades [1,2,3,4]. Only recently, CT type AII was linked to congenital infection with atypical porcine pestivirus (APPV) [2,3,5,6], a pestivirus that was discovered quite recently by metagenomic sequencing of porcine sera within a project on porcine reproductive and respiratory syndrome virus (PRRSV) [7]. APPV is now highly suggested to be the cause of congenital tremor type AII, since clinical signs were shown to be experimentally inducible in gravid sows [2,5,6]. Most often, only the first litter of gilts is affected, whereas the following ones hardly ever show signs of congenital tremor, indicating that the sows develop a protective immunity [2]. In surviving animals, the clinical signs usually disappear within three to four weeks [2,5].

The genus *Pestivirus* within the family *Flaviviridae* comprises bovine viral diarrhea virus 1 (BVDV-1), BVDV-2, border disease virus (BDV), and CSFV, all economically important livestock diseases [8]. The genomic organization of all pestiviruses is very similar and contains an enveloped, positive-sense, single-stranded RNA genome of approximately 12 kb. The genome encodes for approx. 12 proteins in one large open reading frame and the translated polyprotein of around 3635 amino acids is subsequently processed in four structural and at least eight nonstructural proteins. In recent years, several new strains of pestiviruses and new host species have emerged [9], leading to the proposal by the Flaviviridae Study Group of the International Committee for the Taxonomy of Viruses (ICTV) to rename the species independent of their original host. Thus, the classical species were renamed as Pestivirus A (BVDV-1), Pestivirus B (BVDV-2), Pestivirus C (CSFV), and Pestivirus D (BDV), and the currently fully sequenced new isolates as Pestivirus E to K [10]. Finally, new isolates from bats [11,12] and piglets (Linda virus [13]) are yet to be classified. Thus, the atypical porcine pestivirus is currently classified as Pestivirus K.

Since its first description in the US in 2015 [7], the virus has been reported in Brazil [14,15], China [16,17,18], Canada [19], and in several European countries such as Germany [6], Spain [20], Sweden [21], Austria [3], the Netherlands [2], Hungary [22], and in Switzerland in 2017 (C. Bachofen, Institute of Virology, Vetsuisse Faculty, University of Zurich, personal communication; [16]). Hence, it can be assumed that the virus is distributed worldwide at an unexpectedly high prevalence, with approximatively 10% of the samples tested being positive for viral RNA and with a seroprevalence of roughly 60% in most of the studies reported thus far [16,17,20,23,24].

The nucleotide sequence of APPV varies considerably between different strains [24] and is clearly distinct from all other pestiviruses described to date, with nucleotide identities of less than 50% compared to classical pestiviruses [3,7,17]. Recently, another atypical pestivirus was identified in Austria that is most closely related to the Bungowannah virus [13], the latter being described to date exclusively in Australia. This virus was also isolated from piglets with CT, although the animals were reported to display rather lateral shaking of the body, hence the virus was provisionally named as ‘Linda virus’ (lateral-shaking inducing neurodegenerative agent). Currently, the diagnosis of APPV is exclusively done by various RT-PCR assays to detect viral RNA or by ELISA to identify antibody-positive sera based on the few isolates known in each region. Currently, methods to diagnose infections with APPV have not been established in Switzerland, and the types of strains circulating are not known.

Thus, the aim of this study was to determine the type of APPV strains circulating in domestic pigs in Switzerland and its prevalence in a retro- and prospective study by developing a new real-time RT-PCR assay based on published sequences and new sequences from viruses identified in Switzerland. Finally, the possible presence of the Linda virus and any cross-reactivity of the new RT-PCR assay with currently used diagnostic assays for pestiviruses and porcine viruses was investigated, which is especially important considering the freedom of CSFV and the eradication of BVDV in bovines in Switzerland.

## 2. Materials and Methods

### 2.1. Samples

Serum samples were obtained in a monitoring system of the Swiss domestic pig population by the Federal Food Safety and Veterinary Office. The sampling comprised of serum samples up to 2 mL from the years 2011, 2015, and 2018. Furthermore, serum samples from the years 1986 and 2006 were stored at the Division of Swine Medicine at the Vetsuisse Faculty, University of Zürich, Switzerland. Samples from 1986 (*n* = 87), 2006 (*n* = 273), 2011 (*n* = 183), and 2015 (*n* = 180) were collected from adult pigs only in fattening farms, whereas the samples from 2018 (*n* = 357) originated from adult pigs in breeding farms. The clinical history of the animals is unknown. In addition, a few organ samples from suspected clinical cases of CT in piglets were provided by the swine clinic and the Institute of Animal Pathology in Bern in 2017 and 2018 that were used to further characterize the Swiss APPV isolates.

### 2.2. RNA Isolation

RNA isolation of the serum samples was performed following the NucleoMag^®^ VET protocol (Macherey Nagel GmbH, Oensingen, Switzerland), according to the manufacturer’s instructions using the Kingfisher Flex Purification system (Thermo Fisher Scientific, Reinach, Switzerland).

RNA isolation for the organ samples was performed using the Direct-zol™ RNA MiniPrep Kit (Zymo Research, Irvine, USA, cat. no. 2050), according to the manufacturer’s recommendations. Sera and samples were stored at −20 °C before and after RNA isolation, respectively.

As we used serum for the analyses, no housekeeping gene could be used as an internal control, especially to monitor the efficiency of the RNA isolation of each sample. Due to the ubiquitous presence of RNase in these samples, free RNA could not be used as such as control, and therefore, we added Sendai virus in allantois fluid diluted in MEM cell culture medium (Thermo Fisher Scientific), thus representing a control RNA protected from RNase degradation by the virus particle. A defined amount—yielding a final Ct value of approx. 25—was added to each sample prior to RNA isolation. In a separate real-time qRT-PCR reaction, the performance of the RNA isolation and the RT-PCR reaction for potential inhibitors could therefore be evaluated for every individual sample.

### 2.3. Quantitative Real-Time Polymerase Chain Reaction (qRT-PCR)

#### 2.3.1. Primer and Probes for qRT-PCR and Standard RT-PCR

To find optimal regions in the viral genome for positioning the primers and probe, we compiled all available APPV sequences from GenBank^®^ and performed a multiple alignment with the Clone Manager 9 Software (Scientific & Educational Software, Denver, CO, USA). In the conserved 5′UTR region, we designed the primer pair and the probe (Table 1) using the Primer Express Software Version 3.0 (Applied Biosystems, Foster City, CA, USA). The probe was labeled at the 5′-end with FAM and at the 3′-end with minor groove binder (MGB) and the Quencher Q500 (Microsynth AG, Balgach, Switzerland). As new sequences become available over time, the genome alignment was repeated by May 10, 2019 including all available APPV sequences on GenBank^®^ (Appendix A) and all sequences obtained in this study. The alignment confirmed the high conservation of the region within the APPV viral genome selected, and we only detected a minor G/A mismatch in the probe at position 229 (corresponding to sequence accession no. KX929062) in eight out of 50 sequences on GenBank. However, they all belonged to Korean and Chinese isolates, and therefore, no adjustment of the primers and probe or re-analyses of previous samples was required.

As very recently, another new pestivirus in piglets was described in Europe [13], we decided to also screen our sera for Linda virus in addition to APPV. For this new virus genome, only one sequence was available at the time, so we compared it with the sequence of its closest relative, Bungowannah virus. The forward and reverse primer in the 5′UTR was selected in an area where both sequences have a high homology.

For the Sendai virus used as an internal control (see Section 2.2), the primers and the probe were designed according to the published sequence (GenBank accession no. M30202). The probe was labeled at the 5′-end with FAM and at the 3′-end with the Black Hole Quencher (BHQ)-1 (Microsynth AG).

For direct Sanger sequencing of complete viral genomes and partial fragments of the selected isolates, we designed primer pairs covering the whole genome and selected regions (Appendix A).

#### 2.3.2. qRT-PCR

For APPV and Sendai genome detection, a one-step qRT-PCR format was applied. The 4x TaqMan^®^ Fast Virus 1-Step Master Mix (Thermo Fisher Scientific; cat. no. 4444436) was used according to the manufacturer’s recommendations. Briefly, 2 µL of isolated RNA was amplified with 2.5 µL of the 4x TaqMan^®^ Fast Virus 1-Step Master Mix, with 0.4 µM forward and reverse primers and 0.1 µM probe in an end volume of 10 µL. Measurements and analysis were performed using the ABI 7500 Fast Real-Time PCR System instrument and software package (Applied Biosystems, Foster City, CA, USA) using the following thermal profile: 50 °C for 5 min., 95 °C for 20 s, and 45 cycles of 95 °C for 3 s, followed by 60 °C for 30 s. Positive and negative controls where included in every run. For the screening of Linda virus, a SYBR Green one-step qRT-PCR format was used as, based on the limited sequence information available, designing a probe was not feasible. We used the Qiagen QuantiTect SYBR Green OneStep RT-PCR Kit (Qiagen, Hombrechtikon, Switzerland; cat. no. 204243), according to the manufacturer’s recommendations.

#### 2.3.3. Production of Control RNA

##### 2.3.3.1. Linda Virus

Due to the lack of biological material to be used as a positive control for the Linda virus assay, we produced RNA by in vitro transcription in the region of the PCR fragment used. We synthesized a DNA fragment (position 200-367 according to the sequence in GenBank (accession no. KY436034) at Microsynth. This fragment was used to perform PCR (Qiagen HotStarTaq Master, Qiagen, cat. no. 203443), according to the manufacturer’s instructions with the primer at the 5′-end including a T7 promoter sequence (Table 2). The PCR fragment was then loaded on a 2% agarose gel and after electrophoresis, the corresponding band was excised and the DNA purified using the NucleoSpin^®^ Gel and PCR Clean-up kit (Macherey Nagel, cat. no. 740609.50), according to the manufacturer’s instructions. The purified fragment was used to express RNA with the RiboMAX™ Large Scale RNA Production System–T7 Kit (Promega, Dübendorf, Switzerland, cat. no. P1300) according to the manufacturer’s instructions. Briefly, 25 µL of PCR product, 20 µL T7 Transcription 5x Buffer, 15 µL nuclease free water (Ambion; Thermo Fisher Scientific), 7.5 µL of each rNTP’s (25 nM), and 10 µL T7 Enzyme Mix were gently mixed and incubated for 3.5 to 4 h. at 37 °C followed by DNase treatment with RQ1 RNase-free DNase (1U/µg) for 15 min at 37 °C. RNA isolation was performed using the NucleoSpin^®^ RNA Plus Kit (Macherey Nagel, cat. no. 740984.250), according to the manufacturer’s instructions. This in vitro transcribed RNA was purified with the NucleoSpin RNA Plus RNA isolation kit (Macherey Nagel, cat. no. 740984.250), according to the manufacturer’s instructions. The concentration (A260) and purity (A260/280) of the RNA were measured with the NanoDrop™ OneC Spectrophotometer (Thermo Fisher Scientific; Witec AG, Sursee, Switzerland). The RNA was diluted in RNA storage solution (Thermo Fisher Scientific, cat. no. AM7000) and used as a positive control in RT-PCR.

##### 2.3.3.2. APPV, BVDV, and CSFV

To investigate the analytical sensitivity of the new APPV qRT-PCR assay and to test for possible cross-reactivities in the currently applied routine diagnostic assays for ruminant and porcine pestiviruses, we generated in vitro transcribed RNA of APPV, BVDV, and CSFV. For APPV and BVDV, we used the total RNA from an APPV positive sample from Switzerland and from an infected cell culture supernatant with the strain NADL (type strain for BVDV-1 [10]; GenBank accession no. NC_001461), respectively. The PCR fragments with a T7-promoter sequence at the 5′-end (primers, see Table 2) were produced using the OneStep RT-PCR Kit (Qiagen, cat. no. 210212) and the SuperScript™ IV One-Step RT-PCR System (Thermo Fisher Scientific, cat. no. 12594100), respectively, according to the manufacturer’s instructions. The in vitro transcribed RNA was produced as described for the Linda virus in Section 2.3.3.1. CSF viral RNA was kindly provided by Dr. N. Ruggli, and was produced from linearized plasmids of the strains Eystrup (GenBank accession no. NC_002657.1) and Alfort/187 (NC_038912.1) using the MEGAscript™ T7 Transcription Kit (Ambion).

### 2.4. Conventional RT-PCR and Sequencing

To confirm samples with a positive Ct-value in qRT-PCR, we performed RT-PCR with the primer pair APPV-frw-8 and APPV-rev-9 or APPV-rev-49a-d, spanning a larger region in the 5′UTR to N^pro^. If this did not work based on the low amount of virus in the sample, we used the primer pair APPV-frw-8 and APPV-R2-5utr or the primer pair used in the real-time RT-PCR reaction (Table 1 and Appendix A). Amplification was performed on the Veriti™ 96 Well Thermal Cycler (Applied Biosystems) using the OneStep RT-PCR Kit, according to the manufacturer’s instructions. Briefly, 5 µL of RNA was added to a reaction mixture consisting of 9.875 µL nuclease free water (Ambion), 5 µL of 5x Qiagen OneStep RT-PCR buffer, 1 µL dNTP Mix (final concentration of 0.4 mM), 0.125 µL RNasin (40U/µL; Promega), 1 µL of OneStep RT-PCR Enzyme Mix, and 0.6 µM forward and reverse primer in a final volume of 10 µL. The thermal profile was applied as follows: 50 °C for 30 min (RT step), 95 °C for 15 min, 15 cycles of 94 °C for 30 s, 65–50 °C for 1 min (touch down), 72 °C for 1 min/kb, followed by 25 cycles of 94 °C for 30 s, 50 °C for 1 min, 72 °C for 1 min/kb, and finished by 72 °C for 10 min (final extension) and 4 °C as storage step.

The PCR product was purified using the NucleoSpin^®^ Gel and PCR Clean-up Kit (Macherey-Nagel AG), according to the manufacturer’s instructions. Sanger sequencing was performed by Microsynth and the raw data of the sequences where analyzed with the SeqMan™ II sequence analysis software (DNASTAR Inc., Madison, USA).

### 2.5. Phylogenetic Analysis

#### 2.5.1. Phylogenetic Analysis of the Complete Coding Region

Phylogenetic analysis of the complete coding region of the viral genome was performed using a representative sample of all available sequences encompassing the whole genome from GenBank^®^ up to May 10, 2019 (Appendix A). Additionally, the sequences of Swiss origin, newly generated from samples tested positive as described in this study, were included. Codon alignment was performed using MUSCLE [25] included in the MEGA7 software [26]. The evolutionary history was inferred by using the maximum likelihood method [27] based on the general time reversible model with bootstrap values based on 100 replicates. The tree was drawn to scale, with branch lengths measured in the number of substitutions per site. All positions containing gaps and missing data were eliminated. The analysis involved 29 nucleotide sequences, leading to a total of 10,810 positions in the final dataset. Evolutionary analyses were conducted in MEGA7.

#### 2.5.2. Phylogenetic Analysis of a Small Fragment in the 5′UTR

Phylogenetic analysis of the partial genome was performed using a representative sample of all available sequences encompassing the 5′UTR region of the genome from GenBank^®^ up to May 10, 2019 (Appendix A). Additionally, the sequences of Swiss origin, newly generated from samples tested positive as described in this study, were included. The alignment was performed using MUSCLE [25] included in the MEGA7 software. The evolutionary history was inferred using the maximum likelihood method [27] based on the Kimura 2-parameter model [26] with bootstrap values based on 100 replicates. The tree was drawn to scale with branch lengths in the same units as those of the evolutionary distances used to infer the phylogenetic tree. The evolutionary distances were computed using the maximum composite likelihood method [27] and are in the units of the number of base substitutions per site. The analysis involved 94 nucleotide sequences leading to a total of 165 positions in the final dataset. Evolutionary analyses were conducted in MEGA7.

### 2.6. Crossreactivity

To determine the analytical specificity, a dilution series until the negativity of in vitro transcribed RNA of porcine (APPV 5′UTR-N^pro^, CSFV 5′UTR) and bovine (BVDV 5′UTR) origin was produced as described in Section 2.3.3. The RNA was tested in two established (accredited) qRT-PCRs routinely applied for the diagnosis of ruminant pestiviruses and CSFV in Switzerland by the diagnostic division of our institute including one PanPesti qRT-PCR and one CSFV-specific qRT-PCR assay in addition to our newly established qRT-PCR for the detection of APPV in the 5′UTR.

## 3. Results

### 3.1. Development of APPV qRT-PCR Assay in the 5′UTR

To detect the presence of viral RNA, the specific primers and probe designed for the 5′UTR amplification (see Section 2.3.1) were used for qRT-PCR. Positive samples were further confirmed by RT-PCR with appropriate primers, followed by Sanger sequencing. The standard curve of in vitro transcribed APPV RNA placed in the 5′UTR (see Section 2.3.3) was used to determine the analytical detection limit (Figure 1). In six dilution series performed in four independent experiments, 1 × 10^2^ molecules per reaction could be detected in all four experiments with an efficiency of 86.5%.

Considering CSFV to be an important clinical differential diagnosis of APPV, it is of great importance to carefully rule out the possibility of any interference in the established diagnostic tools routinely applied in diagnostics. The in vitro transcribed APPV and CSFV RNA was tested in the qRT-PCR routinely applied in the diagnostics for CSFV when compared to the newly established qRT-PCR for APPV. The RNA was tested in concentrations of 10^5^–10^7^ molecules/reaction for APPV and 7 × 10^5^–7 × 10^7^ molecules/reaction. Larger amounts of viral RNA were not tested (i) to avoid cross-contamination in the routine diagnostic units, and (ii) as none of the field samples in the diagnostics showed lower Ct values than the highest concentration of viral RNA employed. No false positive results were obtained in these qRT-PCR reactions, indicating the specificity of the new qRT-PCR assay for APPV and the currently used test for CSFV. By contrast, occasional interference with Ct values >34 of APPV RNA at concentrations of 10^5^ and 10^6^ molecules/reaction in the PanPesti qRT-PCR was observed, highlighting that cross-reactivity can occur, depending on the assay used. Conversely, no positive result of the in vitro transcribed BVD viral RNA tested in the newly established APPV qRT-PCR was observed, further demonstrating its specificity. Similarly and not unexpectedly, APPV RNA did not cross-react with any assay routinely used in the diagnostic division with porcine sera such as foot-and-mouth disease virus (FMDV), swine vesicular disease virus (SVD), porcine respiratory and reproductive syndrome virus (PRRSV), and African swine fever virus (ASFV).

### 3.2. APPV Prevalence in Switzerland

Previous studies have shown a high virus prevalence in domestic pigs for several different countries including a limited number of samples from Switzerland [16]. In the present study, a representative sample set of 1080 serum samples collected from adult pigs with an unknown history of congenital tremor was tested for the presence of APPV genomes.

For 1986, we detected six positive samples in a set of total 87 serum samples, corresponding to a prevalence of approx. 7% (Figure 2). As these sera were stored at −20 °C for more than 30 years, their quality was severely reduced, and thus, only three samples could be confirmed by sequencing. In 2006, we obtained 50 positive results out of the 273 samples tested, resulting in a high prevalence of 18%. Of all the cantons, the canton of Lucerne showed the highest number of positive samples (17/90 samples; Appendix A) in this year. A similar distribution could be observed for 2011, with a nationwide prevalence of approx. 12% and with most of the positive samples (7 out of 57 samples) again in the canton of Lucerne. In 2015, the overall prevalence remained at a rather high level at approx. 9%. In 2018, sera were obtained from breeding instead of fattening farms, and we could only detect a single positive sample out of 357 sera tested, representing a prevalence of mere 0.3%. Thus, the average viral prevalence in fattening pigs over the years 1986–2015 was high, with approx. 13% on average (Figure 2), and the canton of Vaud had the highest relative prevalence of 33% (Appendix A). In summary, 96 (9%) of the 1080 tested samples were APPV genome positive, ranging from a 0.3% prevalence in 2018 in breeding farms to a prevalence of 7 to 18% in fattening farms in the years 1986 to 2015.

In all of the sera tested from 1986 to 2018, only four had weak positive results in qRT-PCR for the presence of Linda virus. However, all of them were false positive as none of them could be confirmed by sequencing, i.e., the sequences obtained were completely unspecific.

### 3.3. Phylogenetic Analysis

The initial genotyping was implemented based on a small fragment in the 5′UTR of the viral genome. In addition, whole genome sequences of selected samples with a sufficient amount of viral RNA were generated. In addition, four samples from clinical cases of CT in piglets obtained in the last two years were fully sequenced and included in the phylogenetic tree of the complete ORF. Positive samples were further aligned to selected strains available on GenBank^®^. Sequence analysis revealed a broad genetic diversity of all APPV strains. Mostly, no specific clustering dependent on the geographic origin of the strains could be observed, with the notable exception of the Swiss strains. With the small fragment analyzed in the 5′UTR being rather short, the bootstrap values remained generally low in this phylogenetic analysis. However, the branch generating the cluster of Swiss isolates showed a bootstrap value of 66 (Figure 3B), supporting the phylogenetic analysis and cluster formation seen in Figure 3A. Phylogenetic analysis of the complete coding regions and partial sequences in the 5′UTR demonstrated a high genetic diversity of the APPV Swiss isolates, building their own cluster (Figure 3A,B).

Within the Swiss isolates, the results showed no specific spatiotemporal clustering. Thus, the strains were dispersed independently of the year of sampling or their geographical distribution within Switzerland (Figure 3B).

## 4. Discussion

APPV is a recently discovered porcine pestivirus first described in 2015 [7]. Soon thereafter, the virus was linked to congenital tremor type AII in piglets by Arruda et al. [5] and others [2,3,6]. Since then, the virus has been described in a variety of countries, indicating that APPV is distributed worldwide [4,28,29]. In Switzerland, the first report of APPV in a sample from swine was found in a nasal swab analyzed by NGS by C. Bachofen in another project unrelated to pestiviruses (see Section 1). The presence of APPV in Switzerland was confirmed in a study analyzing 120 serum samples from 2015 obtained from fattening pigs, yielding a virus prevalence of 13.3% [16]. In addition, the presence of piglets with congenital tremor of unknown origin have also been reported for many years in clinics in Switzerland. For that reason, we developed a new qRT-PCR assay that also specifically detected the APPV strains isolated in Switzerland and evaluated the presence of the virus in a retro- and prospective study.

Here, we provide strong evidence that APPV has already been circulating in the indigenous pig population for many decades, as indicated by the detection of the virus in samples as far back as 1986. This is, to our knowledge, the most recent report of APPV in pig sera, with the previous report of the earliest detection being reported from 1997 in Spain [20]. The virus prevalence appears to be constantly around approx. 10%, with the observed decrease of the viral prevalence over the years 2006–2015 from 18% to 9% probably occurring by chance based on sampling. This is corroborated by the fact that in the recent study from 2015, a virus prevalence of 13.3% was reported [16], compared to 9% in this study, despite sera from both studies being obtained from the same serum bank of the Swiss Federal Food Safety and Veterinary Office (FSVO). Both studies analyzed samples from all over the country, but as we selected samples according to the distribution of pig farms in Switzerland, the prevalence of approx. 10% over the years might well represent a nationwide distribution of APPV (Appendix A).

Notably, the prevalence in 2018 was only 0.3%. However, the samples obtained prior to the year 2018 were collected exclusively from pigs for slaughter, whereas the samples from 2018 originated from breeding pigs. On the one hand, the awareness of the possible introduction of potential pathogens and the execution of biosafety measures are, for economic reasons, more present in breeding than in fattening farms. On the other hand, and probably more important, is the fact that clinical symptoms of CT only occur in newborn piglets after fetal infection. For the manifestation of CT, it is likely that the infection must occur prior to the development of fetal immune competence (70–80 days of gestation) in piglets [2,3,5]. The clinical signs were reported to disappear over time in surviving animals, leading to the conclusion that the virus will be eliminated. However, the duration of virus elimination in these pigs remains unknown. In addition, it was reported that several pigs with or without clinical signs of CT were still viremic at time of slaughter [2,3,4]. Therefore, the probability of finding virus-positive, older animals in breeding farms is expected to be rather low, as represented by the low prevalence in samples from 2018. As all samples tested in this study prior to 2018 were taken from slaughter pigs at the age of approx. 120–130 days with an unknown history of CT, the high number of samples tested positive revealed a possible chronic infection in these animals. However, we cannot exclude that only a proportion of the animals tested positive for APPV were chronically infected, asymptomatic virus carriers, and that during the vivid re-organization of fattening pigs in different groups of animals of various ages, transient infections by these carrier animals might occur that were similarly detected in our assay at the time of slaughter. Additionally, we could not specifically detect any RNA of Linda virus in any of the sera tested from 1986 to 2018. Thus, there is currently no evidence that the Linda virus might be present in Switzerland.

Interestingly, the Swiss isolates formed their own cluster in the phylogenetic analysis, indicating that they were genetically clearly distinct when compared to the other strains reported worldwide. This signifies that pig farming constitutes a rather “closed” system in Switzerland, with only few imports of live animals. It was proposed that APPV might be transmitted by semen as quite high viral loads were detected in semen or preputial fluid [2,3,30]. Given that a high percentage of sows are inseminated with commercially acquired semen from boar studs, transmission of APPV through semen has to be taken into consideration. Semen is also imported into Switzerland, which might introduce new viruses as reported previously by an outbreak of PRRSV imported by semen from Germany [31]. However, the fact that all APPV strains found in this study were clustered within the “Swiss strains” indicate that either only a few and/or virus-free semen was imported, or that despite the high viral loads in semen, APPV is not readily introduced into a pig herd. The latter has been similarly observed in ruminants, where insemination with BVD- or BD-virus containing semen only sporadically led to persistently infected fetuses [32]. Thus, as all of the isolates analyzed in this study clustered within these “Swiss strains” with no difference in the last 30 years, there is a high probability that APPV has already been circulating exclusively within the Swiss pig population for several decades. This is corroborated by the fact that a similar situation was reported for hepatitis E virus (HEV) infections in Switzerland, with the detection of the proposed genotype 3s being reported only in Switzerland to date [33].

According to sequence alignments, the newly established qRT-PCR should be capable of detecting most of the APPV strains reported worldwide to date. By all means, the assay is highly sensitive for Swiss APPV isolates and therefore represents a highly valuable tool for APPV diagnostics in Switzerland. The latest control of the primer sequences (May 2019) revealed no mismatches in the forward- and reverse primer as well as in the probe for Swiss isolates. However, international strains originating from Asian countries such as China and Korea showed one mismatch in the MGB probe, which signifies that the corresponding primers and probes might need to be adjusted for use in a diagnostic test in these countries. Nonetheless, as rather few APPV sequences are still currently known, continuous sequencing and possible adaption of the diagnostic tests used are highly recommended. Similarly, it is also of great importance to control for any possible cross-reactions within the various diagnostic assays used for porcine samples as APPV needs to be considered an important differential diagnosis to CSF, which is a highly contagious, reportable animal disease with dire consequences for the pig industry if introduced into a herd. Using in vitro transcribed RNA, we could not detect any cross-reactivity of APPV viral RNA in the assays used for the diagnostics of CSFV and other porcine viral diseases. Similarly, CSF viral RNA was not detected by our new APPV qRT-PCR assay, further confirming its high specificity. By contrast, higher amounts of APPV RNA was weakly detected by the “pan-pesti” qRT-PCR applied in our diagnostic division. However, we do not consider this to be a substantial risk even in light of the Swiss BVD eradication program [34], as (i) APPV has never been reported to infect cattle, and (ii) in vitro transcribed APPV viral RNA has been only detected at such high concentrations that we never observed in porcine samples from the field. Thus, these results confirm previous reports [35] that APPV does not interfere with the diagnostic assays routinely applied in the diagnosis of other ruminant and porcine pestiviruses.

In conclusion, the findings of the present study indicate that the newly developed qRT-PCR assay for the detection of APPV in the 5′UTR is able to detect the virus strains observed in Switzerland with high sensitivity and, based on the sequence comparisons, most likely also most of the strains reported worldwide to date. In addition, the retro- and prospective analysis provides strong evidence that APPV has been circulating within the Swiss pig population for more than three decades without any obvious import of virus strains from abroad. As it is unknown whether cross-neutralization of various APPV isolates by antibodies raised against different virus strains occurs, it might be recommended to screen any imports of semen and live animals for the presence of APPV to prevent the introduction of new APPV strains into Switzerland.

## Figures and Tables

**Figure 1 viruses-11-00653-f001:**
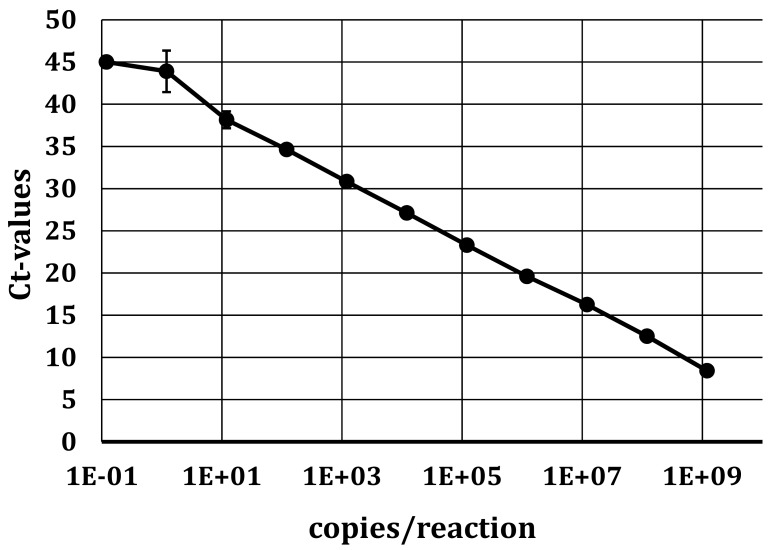
qRT-PCR standard curve of in vitro transcribed APPV RNA in the 5′UTR. A tenfold dilution series within a range of 1 × 10^9^ molecules/reaction to 1 × 10^−1^ molecules/reaction was performed. Targeting the FAM reporter, an efficiency of 86.5% was achieved. The slope of the standard curve was −3.7 with a standard deviation (SD) between 1 and 2.5.

**Figure 2 viruses-11-00653-f002:**
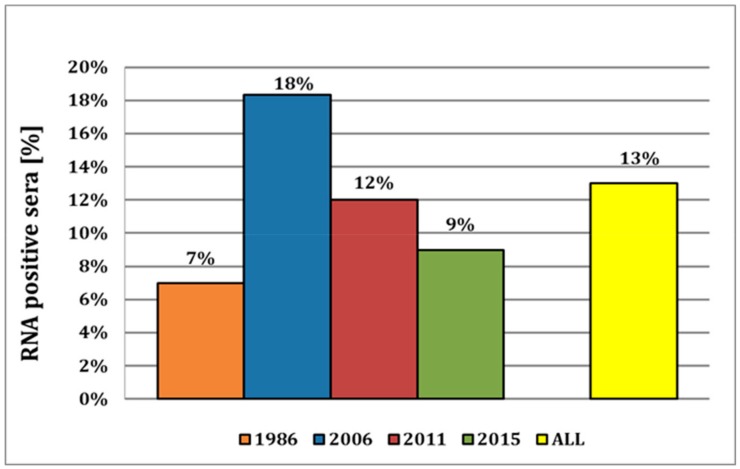
Virus prevalence of APPV in Switzerland from fattening farms. Relative number of APP virus prevalence in Switzerland from 1986 until 2015 with an overall viral prevalence of 13% in fattening pigs.

**Figure 3 viruses-11-00653-f003:**
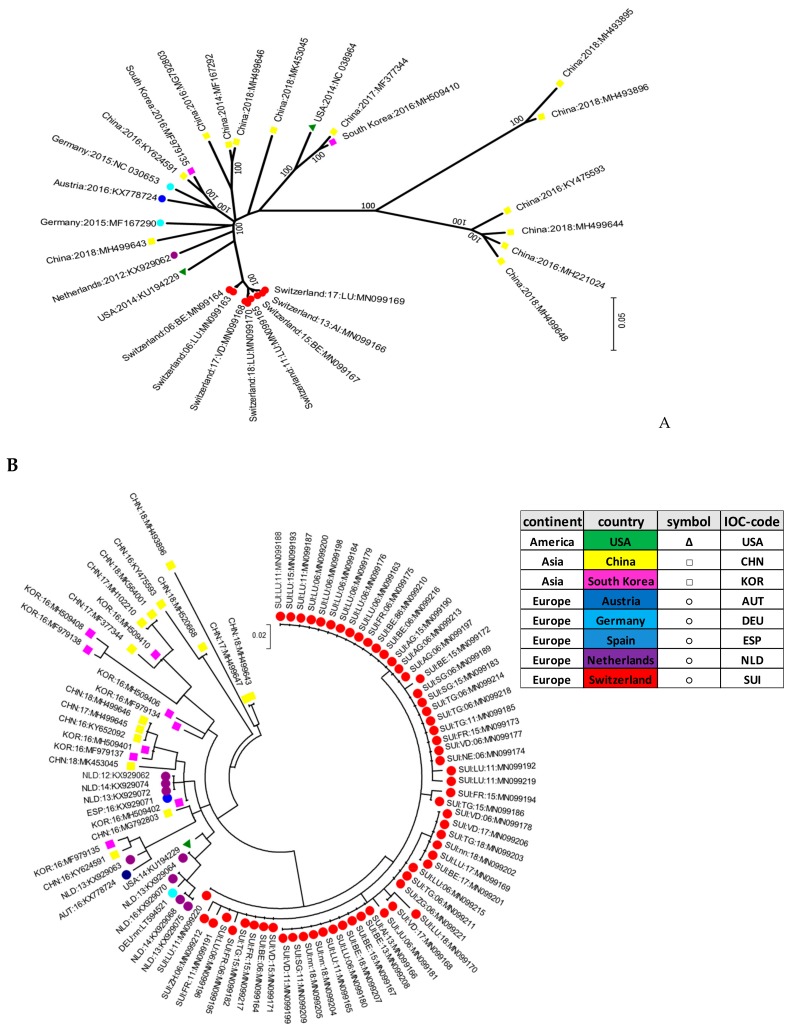
Molecular phylogenetic analysis of the nucleotide sequence of the full ORF of APPV (**A**) or of a fragment in the 5′UTR (**B**). Initial tree(s) for the heuristic search were obtained automatically by applying Neighbor-Join and BioNJ algorithms to a matrix of pairwise distances estimated using the maximum composite likelihood (MCL) approach. A discrete gamma distribution was used to model the evolutionary rate differences among the sites. The trees were drawn to scale, with branch lengths measured in the number of substitutions per site. All positions containing gaps and missing data were eliminated. The percentage of trees in which the associated taxa clustered together is shown next to the branches. In the final dataset, there were a total of 10,810 positions encompassing the full open reading frame (ORF) (A) and 165 positions in the 5′UTR of the virus genome (**B**). Each dot of a specific color was assigned to a specific country including China (CHN) in yellow, South Korea (KOR in pink, USA (USA) in green, Germany (DEU) in light blue, Austria (AUT) in dark blue, Spain (ESP) in middle shade blue, Switzerland (SUI) in red, and the Netherlands (NLD) in purple. The strains are all marked with the IOC-code of each specific country, the year of sample collection, and the accession number. Swiss isolates are additionally marked with the abbreviation code of the canton (see Appendix A).

**Table 1 viruses-11-00653-t001:** Real-time RT-PCR primers and probes.

Virus	Primer	Name	Start 5′	Sequence 5′-3′	bp	Reference Gene
APPV	Forward primer	APPV-F1-5utr	168	GGGCAGACGTCACYGAGTAGTACA	24	KX929062
APPV	Reverse primer	APPV-R2-5utr	340	TCCGCCGGCACTCTATCA	18	
APPV	Probe (MGB)	APPV-MGB3-5utr	214	TGTAGGGTCTACTGAGGCT	19	
Linda	Forward primer	Linda-F1	210	ACCCACTGGCGATGCCT	17	KY436034
Linda	Reverse primer	Linda-R2	337	TCCGCCGGCATCCTATC	17	
Sendai	Forward primer	Sendai-F5	8553	GTCATGGATGGGCAGGAGTC	20	M30202
Sendai	Reverse primer	Sendai-R6	8788	CGTTGAAGAGCCTTACCCAGA	21	
Sendai	Probe	Sendai-P7	8720	CAAAATTAGGAACGGAGGATTGTCCCCTC	29	

**Table 2 viruses-11-00653-t002:** PCR primers for the production of the control fragments (*Bold/italics = T7 promoter sequence*).

Virus	Primer	Name	Start 5′	Sequence 5′-3′	bp	Reference Gene
Linda	Control fragment	Linda-control	200	GGTAAGGATCACCCACTGGCGATGCCTTGT GGACGGGGGCGTGCCCAACGCAATGTTAGC GGTGGCGGGGGCTGCCATCGTGAAAGCTAG GTCTTGATGGACCTTGTTGCCTGTACAGTC TGATAGGATGCCGGCGGATGCCCTGTGA	148	KY436034
Linda	Forward primer (incl T7)	Linda-frwT7-contr	200	ACTG***TAATACGACTCACTATAGGG***AGAGGTA AGGATCACCCACTGGCG	48	
Linda	Reverse Primer	Linda-rev-contr	367	GATATTCTTTATACTGGCTGTCACAGGGCAT CCGCCG	37	
APPV	Forward Primer (incl. T7)	APPV-F8-T7-5utr	130	ACTG***TAATACGACTCACTATAGGG***CTGAGAGAGAGGT ACCGAACTCTTAAG	51	KX929062
APPV	Reverse Primer	APPV-rev-9	890	TCACAATTGGGTTTCCATTGGTA	23	
BVDV	Forward Primer (incl. T7)	BVD386	80	ACTG***TAATACGACTCACTATAGGG***CTCAGCGAAGGCC GAAAAG	43	NC_001461
BVDV	Reverse Primer	BVD387	444	ACCCCGACGGGTTTTTGT	18	

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
