# Peer review of "Long-Term Circulation of Atypical Porcine Pestivirus (APPV) within Switzerland"

_viruses, 2019, doi:10.3390/v11070653_

Reviewer 1 Report

Kaufmann and co-workers report on the detection of APPV in Switzerland in samples obtained from the Federal Food Safety and Veterinary Office of the country from different years. The authors describe a historical and current prevalence of APPV of about 10% in fattening farms and a current prevalence of only 0.3% in breeding farms. They document the prevalence of APPV in Switzerland starting in 1986, and present a phylogenetic analysis of all available APPV genomes indicating an independent Switzerland APPV cluster. Overall the presented manuscript is interesting, clean and the experiments are well controlled. However, it is a pity that serological data on the prevalence of APPV specific antibodies is missing. The detection of animals recovered from APPV infection would be very important for the understanding of APPV epidemiology.

Author Response

We thank the reviewer for his very positive assessment. And we fully agree that serological data would be a rather informative and desirable to obtain. Colleagues, e.g., in Germany (ref 16 in manuscript) or Austria (ref 3 in manuscript), did establish an antibody ELISA that might be used on a collaborative basis. However, as the strains circulating in Switzerland appear to be quite different from all other strains reported to date (this study), we are of the opinion that we need to establish a serological test based on antigens form strains circulating in Switzerland. We are indeed currently developing such an antibody ELISA, but establishing, testing and validating such an assay requires much more time and, thus, the test is not yet deployable.

Reviewer 2 Report

Review of manuscript ID (Viruses-553820) -“Long-term circulation of atypical porcine pestivirus (APPV) within Switzerland

 Manuscript Overview:

This manuscript describes development of a RT-PCR assay targeting the 5’UTR for APPV genome. This assay was developed to retro- and prospectively evaluate the strains present and their prevalence in Switzerland. Of the sera evaluated from 1986-2018, a prevalence of ~13% of pigs at slaughter age and less than 1% of breeding pigs was observed. Positive samples were further evaluated by sequencing to determine the genetic diversity of APPV positive samples. Phylogenetic analysis demonstrate high genetic diversity of APPV isolates in Switzerland with no spatiotemporal clustering.

 Comments:

In general this is a well written manuscript, but the addition of Linda and Sendai for the qRT-PCR actually adds confusion rather than substance. The reasoning is there is really no mention of the lack of detection in the results. Maybe this is supplemental information and just focus on the APPV. To further add to the confusion, control RNA was generated for Linda, APPV, and BVDV, but not for Sendai or CSFV? CSFV does not come up until looking at cross reactivity, but the other pestiviruses were not evaluated for cross-reactivity? Furthermore, NADL was used to generate the control RNA, would you expect NADL to be representative the BVDV diversity and should this have been tested and explained? It is not that there are concerns with the ability of the assay to detect APPV, this is demonstrated by the positive samples being sequenced and confirming they were indeed positive for APPV. The concern is the inconsistency in the controls used or how the controls were used to validate the assay. It would seem that, it could be mentioned that other pestiviruses were evaluated for detection, but it does not need to be a major focus as it detracts from the paper and the take-home message.

Specifically:

Line 20 of the abstract; the beginning of the quote is subscript rather than superscript like the end quote. Otherwise in the introduction line 44 the quote “dancing piglet” appears to be correct.

Author Response

Point 1: (1a) In general this is a well written manuscript, but the addition of Linda and Sendai for the qRT-PCR actually adds confusion rather than substance. The reasoning is there is really no mention of the lack of detection in the results. Maybe this is supplemental information and just focus on the APPV. (1b) To further add to the confusion, control RNA was generated for Linda, APPV, and BVDV, but not for Sendai or CSFV? CSFV does not come up until looking at cross reactivity, but the other pestiviruses were not evaluated for cross-reactivity? (1c) Furthermore, NADL was used to generate the control RNA, would you expect NADL to be representative the BVDV diversity and should this have been tested and explained? It is not that there are concerns with the ability of the assay to detect APPV, this is demonstrated by the positive samples being sequenced and confirming they were indeed positive for APPV. The concern is the inconsistency in the controls used or how the controls were used to validate the assay. It would seem that, it could be mentioned that other pestiviruses were evaluated for detection, but it does not need to be a major focus as it detracts from the paper and the take-home message.

Response 1a: As the issues on Sendai virus and Linda virus, used for technical reasons and as side project, respectively are not the main topic of the article, it might indeed add some confusion to the manuscript. Therefore, we adjusted the corresponding paragraphs accordingly in order to clearly delineate the purpose of the application of these viruses. Thus, (i) the description of the use of Sendai virus was extended in chapter 2.2 (lines 117-126/147) and 2.3.1 (line 147), and (ii) we clarified the aim of the Linda virus PCR in chapter 2.3.1 (line 142-143; compare also response 1b). Finally, we added the (negative) findings of Linda virus in our serum samples in the results section (lines 332-334) in addition to a corresponding change in the discussion (lines 436-440).

Response 1b: In vitro transcribed RNA of APPV, BVDV and CSFV was used to investigate the sensitivity of the new APPV qRT-PCR assay and for their possible cross-reactivities in the routinely applied diagnostic assays for BVDV and CSFV. By contrast, Sendai virus was only used as control in the serum samples (as an RNA protected from RNase degradation by the virus particle, see response 1a) and, thus, no in vitro transcribed RNA was required. However, we indeed omitted the origin of the CSF viral RNA in the methods section by negligence, for which we would like to apologize. We added this description now in section 2.3.3.2, and included the former section 2.3.3.3 into this chapter for ease of reading (lines 189 – 201/220; in addition to adapting the acknowledgements in line 502).

As we were unable to detect any Linda virus in Switzerland, Linda virus in vitro transcribed RNA was exclusively used as positive control in the corresponding qRT-PCR assay (due to a lack of any positive biological material ex vivo or from cell cultures, see section 2.3.3.1), and we were therefore of the opinion, that cross-reactivities with currently applied routine diagnostics tests for ruminant and porcine pestiviruses is not required.

Response 1c: NADL is the type strain of the species BVDV-1 (see reference 10 in the manuscript). We added this information line 194. In addition, decade-long experience with BVDV diagnostics at our institute and sequencing several thousand strains obtained from persistently infected cattle in Switzerland (see Ref. 34 in the manuscript) in the conserved 5’UTR, we are very confident that NADL is indeed representative for these type of qRT-PCR analyses.

 Point 2: Line 20 of the abstract; the beginning of the quote is subscript rather than superscript like the end quote. Otherwise in the introduction line 44 the quote “dancing piglet” appears to be correct.

Response 2: The difference in sub-vs. superscript fonts originates from different computer systems the manuscript was written on, as we could reproduce by re-typing. We corrected the abstract in order to have superscript versions consistently within the manuscript.